# VeVaPy, a Python Platform for Efficient Verification and Validation of Systems Biology Models with Demonstrations Using Hypothalamic-Pituitary-Adrenal Axis Models

**DOI:** 10.3390/e24121747

**Published:** 2022-11-29

**Authors:** Christopher Parker, Erik Nelson, Tongli Zhang

**Affiliations:** 1Department of Pharmacology & Systems Physiology, College of Medicine, University of Cincinnati, Cincinnati, OH 45221, USA; 2Department of Psychiatry & Behavioral Neuroscience, College of Medicine, University of Cincinnati, Cincinnati, OH 45221, USA

**Keywords:** HPA axis, Major Depressive Disorder, stress test, Python, Verification & Validation, differential equations model

## Abstract

In order for mathematical models to make credible contributions, it is essential for them to be verified and validated. Currently, verification and validation (V&V) of these models does not meet the expectations of the system biology and systems pharmacology communities. Partially as a result of this shortfall, systemic V&V of existing models currently requires a lot of time and effort. In order to facilitate systemic V&V of chosen hypothalamic-pituitary-adrenal (HPA) axis models, we have developed a computational framework named VeVaPy—taking care to follow the recommended best practices regarding the development of mathematical models. VeVaPy includes four functional modules coded in Python, and the source code is publicly available. We demonstrate that VeVaPy can help us efficiently verify and validate the five HPA axis models we have chosen. Supplied with new and independent data, VeVaPy outputs objective V&V benchmarks for each model. We believe that VeVaPy will help future researchers with basic modeling and programming experience to efficiently verify and validate mathematical models from the fields of systems biology and systems pharmacology.

## 1. Introduction

The life cycle of a computational model involves development, verification, validation, and application. Before a model can be confidently applied to help solve a problem, it must be carefully examined and evaluated. The process of evaluating a model includes two steps: verification and validation (V&V). According to Thacker et al. [1],
“*Verification is the process of determining that a model implementation accurately represents the developer’s conceptual description of the model and its solution. Validation is the process of determining the degree to which a model is an accurate representation of the real world from the perspective of the intended uses of the model*.” 

In certain fields of mathematical modeling, such as nuclear engineering, model V&V has been performed following well-defined procedures for decades [2,3]. In the 1980s, for instance, the International Atomic Energy Association defined standardized benchmarks for validation of models of reactor cores [2]. In a paper from 1993, Nakagawa applies the benchmarks to prove the validity of their model of a reactor core [3]. The practice of creating standardized benchmarks for V&V has persisted in nuclear engineering—as seen in a paper by Höhne et al. from 2018 [4]. The standardization of V&V procedures is not limited to nuclear engineering, and can be found in other fields of engineering, as well. For instance, the American Society of Mechanical Engineers published a set of standards for V&V in computational solid mechanics and the American Institute of Aeronautics and Astronautics also published a set of standards for computational fluid dynamics [5].

However, V&V practices in systems biology and systems pharmacology are still being improved to meet significant challenges, in part due to the individual variability and resultant complexity inherent to physiological systems [6,7]. For example, Hicks et al. [8] present best practices for V&V of neuromusculoskeletal modeling and the basic concepts presented are applicable for most models in systems biology and systems pharmacology. For instance, “creation of gold standard datasets” and ensuring that efficient tools for V&V are available are excellent goals for the physiological modeling research community, in general. In this work, we have followed these recommendations and tailored some of the specifics to meet the needs of HPA axis modeling. 

In systems biology and systems pharmacology, the ideal model would describe a physiological system adequately in any situation—including exposing the system to a variety of inputs, such as stress or pharmaceuticals. In practice, it is difficult to develop a systems biology or systems pharmacology model that is generalizable to situations even slightly different from the original research. A practical challenge for many researchers using mathematical models is to quickly and efficiently determine which model from the literature is best suited to their current work—or which model could be most effectively modified to fit their needs. Unfortunately, the lack of useful tools for V&V in the field often means that developing a new model from the ground up is more efficient than constructing and testing models from the literature in search of a viable candidate.

### 1.1. Custom Tools to Facilitate Model V&V

In order to help future researchers carry out V&V efficiently, we have developed a Python code library, VeVaPy, with several useful modules for this purpose. The level of difficulty of model V&V represents a significant gap in the field of HPA axis research, one that we aim to fill with our tools and data. Currently, it requires a high level of programming expertise to take a model from the literature and reconstruct it for V&V—the tools available for this purpose (at least for non-stoichiometric models) are not designed for ease of use by biologists. We concede that our V&V code still requires a degree of programming know-how at this point, but we believe that it is a significant improvement over the status quo.

The four modules of the code library are called dataImport (includes several HPA axis data sets for use in model validation, with ACTH & cortisol concentration data at rest and under acute stress), DEsolver (more streamlined differential equation solver, works with ODE or DDE systems), optimize (easily facilitates parameter optimization), and visualize (generates graphs based on user specifications). We use VeVaPy to demonstrate several case studies of HPA axis model V&V—similar to the case studies presented by Hicks et al. [8].

### 1.2. Validation against Novel Data Collected in MDD Patients 

For validation of the HPA models we present as case studies, we compare them against new and independent data collected from Major Depressive Disorder (MDD) patients undergoing stress tests. MDD is a mental disorder with severe implications for quality of life. Symptoms include weight loss/gain, hypersomnia or insomnia, slowing of speech and action, impaired concentration, depressed mood, decreased interest in work/hobbies, low self-esteem, increased feelings of guilt, and suicidal thoughts [9]. For depression to be considered MDD, the symptoms must last a minimum of two weeks and cause significant difficulties functioning at work and interacting socially [9]. There are three main subtypes of MDD with significant differences in symptoms: melancholic depression is characterized by weight loss and insomnia, atypical depression is characterized by weight gain and hypersomnia, and uncategorized depression does not fit neatly into either of those two subtypes [10]. The differences between subtypes likely extend beyond symptoms, with some authors hypothesizing that different physiological features are associated with each subtype of MDD [10,11].

MDD is linked to multiple types of physiological disruptions, for example, neuroimaging features or sleep EEG disturbances [11]. However, we are primarily interested in the link between MDD and dysregulation of the hypothalamic-pituitary-adrenal (HPA) axis. There is a lack of consensus regarding whether MDD subjects generally exhibit HPA axis dysfunction—melancholic MDD subjects, however, are more likely associated with increased HPA axis activity and hypercortisolemia [10].

The HPA axis is a neuroendocrine system involved in the body’s stress response. On exposure to a stressor, the paraventricular nucleus (PVN) of the hypothalamus releases corticotropin-releasing hormone (CRH). CRH is released not into the systemic circulation, but into the hypophyseal portal system connecting the hypothalamus directly to the anterior pituitary [12]. The anterior pituitary releases adrenocorticotropic hormone (ACTH) into the systemic circulation in response to increased CRH concentration. The main target of circulating ACTH is stimulation of glucocorticoid production/secretion in the zona fasciculata of the adrenal cortex [12]. The glucocorticoid synthesized is cortisol in humans and corticosterone in rodents. During this process, very little CRH enters the systemic circulation (making collection of CRH concentration data exceedingly difficult), while levels of ACTH and cortisol are readily detectable in blood.

Cortisol acts on various tissues throughout the body by way of glucocorticoid receptors (GRs)—which are nearly ubiquitous—and mineralocorticoid receptors (MRs). Both receptor types translocate to the cell nucleus when bound to cortisol and exert their effects through stimulation or repression of gene transcription. The stress response generated by cortisol includes immune system suppression, increased gluconeogenesis, and increased metabolism of fat, protein, and carbohydrates. Another important function occurs in the hypothalamus and pituitary as cortisol-GR binding decreases the synthesis of CRH and ACTH, respectively. In this way, cortisol exerts negative feedback on its own production. However, GR binding in the hippocampus serves to stimulate CRH production, so the system has both positive and negative feedback mechanisms to consider [13].

Concentrations of cortisol and ACTH normally exhibit both circadian and ultradian oscillations. Circadian refers to the oscillations with a period of roughly 24-h. These oscillations are largely controlled by the circadian clock in the suprachiasmatic nucleus (SCN) of the hypothalamus. However, many peripheral tissues contain lesser circadian clocks, including the adrenal glands [13]. The circadian oscillation of cortisol and ACTH peaks around 8 AM, decreases until after midnight, and then increases again until the morning peak. Ultradian oscillations have a period of 60–90 min and represent 12 to 18 episodes of cortisol/ACTH secretion throughout a day, with little to no secretion between them [14]. Both forms of oscillation likely exist to facilitate more rapid and stronger stress reactions at certain times of day. It has been shown that responses to noise stress are reduced during non-secretory periods and increased during secretory periods in rats [13].

If cortisol levels are sustained at high or low levels for too long, the health consequences are typically serious. Hypercortisolism is a chronic elevation of cortisol concentration, and it is implicated in the development of depression, cardiovascular disease, and Type 2 diabetes mellitus [13]. Hypocortisolism is a chronic decrease in cortisol concentration that is associated with impaired memory formation and post-traumatic stress disorder (PTSD) [13]. Several authors suggest that hypocortisolism is likely caused by increased negative feedback of cortisol on the HPA axis while hypercortisolism is likely caused by decreased negative feedback of cortisol on the HPA axis [15,16]. According to Holsboer [16], the negative feedback caused by cortisol binding GRs is impaired in MDD, likely due to decreased sensitivity and density of GRs.

Treatment options for MDD patients include evidence-based psychotherapies such as cognitive behavioral therapy (CBT) and/or antidepressant medications such as selective serotonin reuptake inhibitors (SSRIs). According to Holsboer [16], in patients with depression, elevated CRH levels lead to hypercortisolism. SSRI therapy is associated with normalization of CRH and cortisol concentrations in these patients [16] possibly by inducing an upregulation of MRs, which are known to inhibit HPA axis activity [17].

If the hypothesis of Holsboer [16] holds and MDD patients (or at least melancholic MDD patients) have decreased sensitivity and density of GRs, we should be able to detect differences in the behavior of the HPA axis under stress when compared with healthy controls. We would expect to see the concentration of CRH, ACTH, and cortisol rapidly increase on exposure to a stressor—the same process we would see in a healthy subject. However, with cortisol at a high concentration, we hypothesize that we would see a slower return to basal concentrations in MDD patients because diminished GR activity would result in decreased inhibition of CRH and ACTH secretion. The best way to test this hypothesis is to take blood samples and measure ACTH and cortisol at short intervals while MDD patients and healthy controls experience a significant amount of stress. Our chosen method of producing a stress response in a laboratory setting is the Trier Social Stress Test (TSST)—see Section 2.3 for a description of TSST procedures.

### 1.3. Model Validation against Experimental Data

A model that can accurately simulate the HPA axis during a TSST will allow us to make predictions about differences between MDD patients and healthy controls, so our validation procedure for the case studies presented herein is based on their ability to match TSST data. The state of HPA axis modeling in the literature is described in the following section, and the procedure by which we chose models to use as case studies is described in Section 2.1. After model selection, we perform our verification procedure (described in Section 2.2) and our validation procedure. 

For validation, we begin by running a parameter optimization algorithm on each model. This yields the optimal parameters for matching stress test data (optimal parameters are generated for data sets from seven patients and the mean concentrations of all 58 patients). To compare model data matching objectively and quantitatively for the TSST data, we compute a cost function for each model on each data set. The result of the cost function is used by the parameter optimization algorithm to determine the suitability of parameter sets, comparing many sets to each other until it finds the optimal parameters for each model on each data set. We also use the cost function to assess the validity of models, as it indicates how well the model has simulated the experimental situation. Furthermore, the cost function values allow us to compare models to each other, and this allows us to objectively determine which model fits our data sets most closely. For a discussion of how the cost function is computed, see Section 2.4.

### 1.4. Mathematical Models of the HPA Axis in the Literature

There are many mathematical models of the HPA axis in healthy, MDD, and PTSD subjects in the literature. These models are predominantly ordinary differential equation (ODE)-based, although there are also some delay differential equation (DDE)-based models. They primarily vary in the way circadian oscillations are generated and (if they are considered at all) how ultradian oscillations arise.

Figure 1 shows a timeline of HPA axis modeling, starting with the first ODE model of an oscillating biological system by Goodwin in 1965 [18]. This model included a negative feedback loop to produce oscillating solutions rather than the steady-state solutions seen in prior models. While not specifically modeling the HPA axis, this was a direct precursor to the modern form of HPA axis models. The first improvement was made by Veldhuis et al. in 1989 with an HPA axis model attempting to model the ultradian rhythm of cortisol [19]. However, this model was a convolution model rather than an ODE model. In 1994, Gonzalez-Heydrich et al. published an ODE model including equations for CRH, ACTH, and cortisol [20]. This was the first “modern” model of the HPA axis, and the basic structure of models has remained largely the same since. The model by Liu et al. in 1999 was an ODE model with five equations, including CRH, ACTH, free cortisol, cortisol binding globulin (CBG)-bound cortisol, and albumin-bound cortisol—CBG and albumin are the two main proteins that bind and inactivate cortisol in the blood [21]. It was able to produce ultradian oscillations in CRH, ACTH, and cortisol concentrations, but failed to produce circadian oscillations.

The remaining models on the timeline are the five models we reproduce for this paper, plus the model by Andersen et al. [27] that we have included in the Appendix A (due to a lack of valid figures for verification in the model paper). First, in 2008, Bairagi et al. produced a DDE-based model that included delays between the production of ACTH/cortisol and their action [22]. The model was able to produce both ultradian and circadian oscillations but required a pulse generator function representing the suprachiasmatic nucleus (SCN) of the hypothalamus for the circadian oscillations. In 2012, Sriram et al. published a model with four ODEs, including CRH, ACTH, cortisol, and GR availability/binding [23]. The inclusion of GR interactions allowed this model to produce both circadian and ultradian oscillations without external input from the SCN. The model by Andersen et al. in 2013 was a DDE-based model, and the authors attempted to produce oscillations by introducing hippocampal GR/MR interactions [24]. However, this model was unable to produce any oscillations for physiologically reasonable parameter values. In 2015, Malek et al. published an ODE model of the HPA axis and its interactions with inflammatory cytokines [25]. The model also ran as an HPA axis-only model by setting the initial concentrations of the cytokines to zero and was able to produce both types of oscillations desired, circadian and ultradian (through an external pulse generator function). Bangsgaard & Ottesen published an ODE model in 2017 that innovated by matching experimental data from individual patients with a parameter optimization procedure [26]. This allowed the authors to determine differences in parameters between healthy control subjects and depressed subjects. Finally, in 2020, Somvanshi et al. published another ODE model of the HPA axis and its interactions with inflammatory cytokines [27]. Their model differed from that of Malek et al. because it included equations for many other species, including GRs.

## 2. Methods

We used several tools to ensure that VeVaPy is easily accessible, well documented, and user-friendly, for the convenience of future researchers attempting to perform mathematical modeling of the HPA axis. The code for the demonstration models was written in Jupyter notebooks (https://www.jupyter.org, accessed 23 November 2022), which allows for Markdown text in between code segments. This allowed us to include a Table of Contents in each notebook for easy navigation, as well as well-formatted and easily readable instructions for use of the models. These Jupyter notebooks are all publicly available on Github (https://www.github.com/cparker-uc/VeVaPy, accessed 23 November 2022) and can be run on any computer with internet access and a web browser through Binder (see our Github repository for instructions; Binder homepage: https://www.mybinder.org, accessed 23 November 2022). We present further information about these tools in the Discussion. 

### 2.1. Model Selection

We selected models for demonstration of VeVaPy by searching PubMed for “HPA Axis Mathematical Model”, on 26 August 2021. This search yielded 1023 results. We selected all papers which included language in the abstract suggesting that a mathematical model was used to study some feature of the HPA axis, a total of 41 papers. Then, an initial screening analysis was performed on all search results, which eliminated all papers not related to differential equations modeling of the HPA axis—35 papers remained at this point. We then performed a more thorough analysis of the remaining papers, and selected all which met the following set of criteria:Included all necessary equations in dimensional form, 16 models selectedIncluded all parameter values used for at least one figureNot substantially similar to an earlier modelThe model is based on the human HPA axisWe have excluded our own models

Five models that satisfy all of these criteria are deemed to have high potential for successful V&V and further tested in our work. 

### 2.2. Model Verification

We modified the general VeVaPy model template for each of the five selected models, inputting the systems of differential equations, parameter values and bounds, and initial conditions. To verify that the models were performing as the authors intended, we ran simulations to recreate figures from the model papers. This process requires the most modification of the template, because each paper includes very different figures that must be replicated. While we have developed the visualize module of VeVaPy to create graphs of variable concentrations over time, we are still working on expanding it to accommodate different formats (for instance, multiple variables on the same graph, as shown in the third figure of Malek et al. [28], see Appendix A). The results of this process are discussed in Section 3.4.

### 2.3. Data Collection 

The data VeVaPy used includes new patient data locally collected and data that are electronically collected from previous publications. For electronic collection, we used LabNotes software (http://mpf.biol.vt.edu/lab_website/Labnotes.php, accessed 23 November 2022), and the filenames indicate the data sources as follows: Bangsgaard-Ottesen-2017 [26], Bremner-2007 [28], Carroll-2007 [29], Golier-2007 [30], Yehuda-1996 [15]. These data represent basal concentrations of cortisol or both cortisol & ACTH measured at short intervals over 24 h. The other data included in VeVaPy come from patients undergoing TSST as described below.

The data were collected following clinical research procedures approved by the IRBs of University of Cincinnati and Cincinnati Children’s Hospital. Briefly, subjects were initially screened with the Structured Clinical Interview for DSM-IV (SCID) and the Inventory of Depressive Symptoms clinician-rated version (IDS-C) by a trained clinician. A total of 88 subjects between the ages of 18 and 65 were selected for the study, with 22 being healthy controls and the other 66 fulfilling the following criteria: DSM-IV criteria for a major depressive episode, either meeting the modified criteria listed in Appendix A for melancholic or atypical depression, or not falling in any depressive subtype; a score on the IDS-C of 20 or greater. Several exclusion criteria were also defined, as listed in Appendix A. All subjects were given an opportunity to read the informed consent document and the protocol was verbally explained at the screening visit. This procedure was approved by IRB review.

Subjects returned at 5:00 PM on the first day of testing and stayed at the General Clinical Research Center (GCRC) of Cincinnati Children’s Hospital until all testing was completed at 6:00 PM on the third day. Blood samples were collected at 10-min intervals to determine basal levels of cortisol and ACTH from 8:00 PM to 9:00 PM on day 1 and 8:00 AM to 9:00 AM on day 2. Subjects also had saliva samples taken every 20 min during these time intervals to serve as a measure of free cortisol. 

A Trier Social Stress Test (TSST) was performed on the second day starting at 5:00 PM. The test involved subjects making an oral presentation to a panel of judges (whom the subjects were told were scientists specializing in behavior analysis), ostensibly to convince the judges that they are the most qualified candidate for a job opportunity related to their interests. Following the oral presentation, there was a question/answer session with the judges and then the subjects were given a series of mental arithmetic tasks to perform for the next five minutes. The total time for the presentation, question/answer, and mental arithmetic tasks was 20 min. Blood samples to determine cortisol and ACTH levels were drawn 30 min and 15 min before the TSST began, at the beginning of the TSST, 10 min and 20 min into the TSST, and then every 15 min for 90 min after the conclusion of the TSST. Heart rate was also measured during the TSST, and saliva samples were collected at the start of the TSST, the end of the TSST, and every 30 min for 90 min after the conclusion of the TSST. Note that, following the TSST, all subjects were informed that the panel of judges were not actually scientists and had been instructed to not react or offer positive feedback during the presentation.

The subjects also underwent a combined dexamethasone-CRH (DEX/CRH) test. This began with 1.5 mg of dexamethasone administered at 11:00 PM on day two. Saliva and blood samples were taken at 8:00 AM on day three to determine cortisol, ACTH, and dexamethasone levels (dexamethasone levels to control for differences in dexamethasone metabolism). At 2:30 PM and 3:00 PM on day three, the subjects once again had blood drawn to determine basal levels of cortisol and ACTH before the CRH test. Then, at 3:00 PM on day three, the subjects were administered 100 mcg ovine CRH (oCRH). Blood samples were taken every 15 min for the first hour and every 30 min for the second and third hours to determine cortisol and ACTH levels. Saliva samples were collected every 30 min for three hours following dosing with oCRH. This concluded the procedure, and the subjects were dismissed.

To facilitate matching the cortisol and ACTH concentration data, we excluded subjects with any data points missing. There were a total of seven MDD subjects and one control subject lacking at least one data point, so overall we had 58 subjects to use for modeling purposes. All subjects used in our modeling for this paper underwent the TSST, and the subjects included 43 diagnosed with MDD and 15 healthy control subjects. We have not included any analysis of the DEX/CRH data in this paper, however, this data will be useful for future analyses using a model modified to allow dosing with dexamethasone and oCRH.

### 2.4. Model Validation

Each model was run with a parameter optimization algorithm against a subset of the patients from the TSST data set and the mean cortisol and ACTH concentrations between all patients. We did not perform this process against all 58 patients due to the extreme time and computational power requirements of such an undertaking. We used the scipy.optimize.differential_evolution package for parameter optimization. The cost function used for parameter optimization involved creating splines between simulated points for ACTH and cortisol and computing the mean sum of squared errors between the splines and the data to be matched. The equation is as follows:$$cost=\frac{\sum_{i}{\left(d_{ACTH,i}-s_{ACTH}\left(t_{i}\right)\right)}^{2}+{\left(d_{CORT,i}-s_{CORT}\left(t_{i}\right)\right)}^{2}}{2}$$
where $d_{ACTH,i}\;\&\;d_{CORT,i}$ are the data points at time $t_{i}$, and $s_{ACTH}\left(t_{i}\right)\;\&\;s_{CORT}\left(t_{i}\right)$ are the spline functions for the simulated ACTH and cortisol, respectively, at time $t_{i}$. The splines’ points were normalized to the mean concentration of the respective data set to be matched, and the data sets were normalized to their mean, as well. This normalization procedure allowed us to compare cost function values between models, even when the models operated on different time/concentration scales. 

The reason for creating splines between the simulated points when computing the cost function is to handle a limitation in the ODE solver methods available in Python (and MATLAB, also, because the same differential equation solver method is commonly used in both languages). The problem is that the step size of the solver is not fixed, so we cannot guarantee that we will have a solution at the exact time point in the data being matched. Although we may get very close, the time steps are very often off by a small amount. The best solution we have found is to compute splines between each point in our solution array, and then select the points on the splines to exactly match the time points of data.

Each model was run against the concentrations averaged over all patients because it is the best example of how we expect the concentrations to behave (starts low before the stress test, peaks during and shortly after the test, returns quickly to baseline before measurement period ends). To illustrate the differences observed in individual patients, the models were also run against several individual patients (patients 1, 10, 20, 30, 40 & 50). Of particular interest, we demonstrated the results against patient 1 as an example of the data sets that fit into neither our understanding of how ACTH and cortisol concentrations should interact with each other nor how they should behave after exposure to a stressor (ACTH is decreasing over nearly the entire measurement period, yet cortisol spikes 30 min after the TSST ends). We also illustrate the model simulation results against patient 40 as a good example of how the concentrations should behave (with limited individual variation, making it distinct from the mean data set). The processes were facilitated by the ability of VeVaPy to efficiently plug in different data and model, as described below.

We tested using alternative cost functions and optimization algorithms and chose the mean sum of squared errors for the cost and differential evolution for the algorithm because they outperformed the other options. The alternative cost functions we tested involved using the maximum of the maximum distances between the ACTH & cortisol simulation spline curves and the real-world data, or the mean of the maximum distances between ACTH & cortisol simulation spline curves and the real-world data. These cost functions performed slightly worse overall when compared to the mean sum of squared errors. For alternative optimization algorithms, we tried using scipy.optimize.shgo and scipy.optimize.basinhopping. These algorithms performed worse than differential evolution, in general. However, all of these methods can be implemented easily in VeVaPy, by passing different arguments to the optimize module.

When choosing which parameters to optimize for each model, we considered the authors’ intentions and tended to optimize only those parameters which were reported to vary between patient populations, at first. However, to demonstrate the maximum effectiveness of parameter optimization, we also ran simulations where we optimized every parameter (simply to determine the optimal data matching from each model). However, we did not modify the equations for any of the models used for demonstration. The only changes made from the original model publications is the parameter values we have optimized.

## 3. Results

First, we present a simplified description of how VeVaPy functions, leaving all technical information to Section 3.1. The tool contains a template that can be edited to add information about the model to be simulated. The information required for VeVaPy to function with a novel model includes: the system of differential equations constituting the model, the parameter values (and reasonable bounds on those parameters for optimization), and initial conditions for each variable (although these can also be optimized given reasonable bounds). The tool then uses the enclosed modules to simulate the system and optionally optimize parameters—the modules will be described in detail in the next section. The outputs from VeVaPy include the optimized parameters, the quantitative description of data matching suitability and graphs for visualization of the simulations. Figure 2 shows the simplified input/output diagram of VeVaPy.

### 3.1. Code Diagram and Module Descriptions of VeVaPy

See Figure 3 for a diagram of the Jupyter notebook template used for each of the five demonstration models. The following section describes each step in the template in turn (following the numbering seen in the left-hand block of the diagram).

The VeVaPy template begins with Section 1. Parameter Definitions, where parameters are defined, along with bounds on each parameter for use in optimization. In this section, initial conditions (or bounds on initial conditions for optimization), time scale, and integration time length are also defined. Section 2. Import Real-World Data and Graph calls the dataImport module using the time scale defined by the user. The module imports all of the data we have gathered on the HPA axis (both basal data and TSST data) into arrays for analysis. This makes validation more streamlined and more powerful, as one can access a wide range of data sets without needing to scour the literature. We have also included code in the template to plot each data set from dataImport to allow users to easily see differences between data sets. 

In Section 3. Optimization Loop, the parameter optimization is performed by the optimize module. The user adds the system of equations for their model into the model function (in the ode_system subfunction). Then, the user calls the run() method of the optimize module. This sets up and runs the parameter optimization algorithm, which repeatedly calls the model function, passing a set of parameters each time. 

The model function then calls the DEsolver module using the system of equations defined by the user and the parameter set from the optimization algorithm. DEsolver allows for solving ODE and DDE systems in a user-friendly fashion. Currently, it is not straightforward to solve ODE systems in Python when using any solver other than the default (lsoda from the FORTRAN library odepack), and we are unaware of any straightforward methods for solving DDE systems. VeVaPy makes both of these possible for HPA axis models, requiring only a function defining the equations and a single call to the module. 

DEsolver then returns the solution of the system, which is passed by the model function to the optimization algorithm, in turn. The algorithm then calls the cost function, passing this solution array. The cost function then calls a cost() method of the optimize module, depending on which cost function is desired (for instance, sum of standard errors is SSE_cost). This method takes arrays containing simulated data and the real-world data to use for validation and returns a single value for cost—representing the suitability of the parameter set tested. Based on this cost value, the optimization algorithm determines the most accurate parameter set that it can find, and then this set is saved and the loop repeats. 

After the loop has run as many times as desired, Section 4. Save Optimization Results to File runs, and all optimized parameter sets and solution arrays are saved to an Excel file. Finally, in Section 5. Plot Optimized Simulations Against Real-World Data, the VeVaPy module visualize is used for creating graphs of each variable comparing the simulated values with the data set used for validation. 

Note that when performing verification, the optimization loop is replaced with a single call to the model function, which then uses DEsolver to solve the system using the parameter values provided. The template contains code for this purpose, with modification necessary only to ensure that the graphs generated contain the same information as those the model is being verified against.

Not only are these modules useful for reproducing HPA axis models from the literature, but they can also be used for creation of new models of the HPA axis or potentially generalized to model other systems. Given the extensive experience and knowledge typically required to create a differential equation-based model of a physiological system, or even to reproduce one from the literature, we have attempted to make VeVaPy as user-friendly as possible to allow a broad audience to use it. As explained at the beginning of Section 2, all five of the models have been written in Jupyter notebooks with thorough documentation explaining the purpose of each code segment. Furthermore, the use of our custom library has been demonstrated in these models, and the code for each module in the library also includes thorough documentation. As a result, the reproduction of an HPA axis model starting from our template Jupyter notebook will be much more easily accomplished than starting from scratch.

### 3.2. Description of Collected Data

The data used for our validation demonstration are described below. As seen in Figure 4, the mean concentrations of ACTH and cortisol from the MDD patients before, during and after administration of a TSST follow the expected trend. Levels are steady and comparable to basal concentrations of MDD patients during the 30 min leading up to the test. During the 20 min that the subjects were participating in the TSST, levels sharply increase and then decrease back to baseline over the 90 min following the end of the test.

However, there is a large amount of variation between subjects. Of the six subjects chosen for matching, the general trend of an increase in ACTH and cortisol concentration followed by a decrease back to baseline is observable in patients 10, 20, 30, 40, and 50—although the degree to which concentrations increase and decrease varies widely. As an illustration of this point, the data for patient 1 and patient 40 are shown in Figure 5A,B, respectively. Strangely, patient 1 exhibits the largest peak in cortisol concentration at 18:20, 30 min after the conclusion of the TSST. Additionally, the ACTH concentration data for patient 1 is decreasing over nearly the entire time frame, which does not coincide with the increasing cortisol concentration. Therefore, it is to be expected that mathematical models will struggle to match the data from this subject. We expect that the mean concentration data, along with patients 10, 40, and 50 will be most successfully matched as they most closely follow the expected trend.

### 3.3. Summary of Selected HPA Axis Models

The five models selected following our search of the literature include: Bairagi et al. [22], Bangsgaard & Ottesen [26], Malek et al. [25], Somvanshi et al. [27], and Sriram et al. [23]. See Table 1 for a summary of the characteristics of each model—including the number of equations, number of parameters and number of feedback loops (all of which give some indication of the amount of detail included in the system). Two of the papers were primarily interested in whether the HPA axis system itself exhibited ultradian oscillations or whether clock inputs from the brain were necessary, two of the papers used their models to study the interactions between the HPA axis and inflammatory cytokines, and the final paper was interested in determining whether PTSD patients exhibited stronger negative feedback from cortisol on the hypothalamus and pituitary than control patients.

Two of the models [23,27] replace the concentration of cortisol with the concentration of bound GRs when computing negative feedback—which is logical because cortisol must bind GRs to exert its negative feedback. In the model by Sriram et al. [23], this allows for the introduction of a positive feedback loop in the receptor binding equation, which generates bistability and therefore Hopf bifurcations in the model (which is an indication that the model can successfully generate ultradian oscillations without needing an external pulse generator function) [31].

Aside from the differences in handling negative feedback interactions, the other major difference in the models is the presence or absence of a function modeling external circadian drive from the SCN. Four of the models include a function for the SCN drive in the equation for CRH [22,25,26,27] while the other model does not include any circadian drive input from outside the HPA axis [23]. The model by Somvanshi et al. [27] also includes a function in the equation for ACTH to describe the adrenal circadian clock drive.

### 3.4. Verification of Selected Models with VeVaPy 

To verify that each model performs as the authors intended, we reproduced a figure from each original paper. All of the model papers contain at least one figure in which cortisol concentration over time is shown—these are the figures we have reproduced. Figure reproductions and the original figures are included in Appendix A. 

In order to generate the figures, the VeVaPy template is edited to include the system of model equations, the parameter values and initial conditions defined in the model publication, and the time over which to integrate. The VeVaPy module DEsolver then solves the system and returns a solution array to visualize, which generates figures that are comparable to the publications. The module visualize allows users to define the variables to plot and the ranges over which to plot them. For each variable, a graph is produced showing the concentration values over the requested time range. The graphs can contain both simulation results and real-world data, but only one variable can currently be shown per graph. As mentioned in Section 2.2, this cannot accommodate all figures for verification currently (such as those in Malek et al. [28]). However, many verifications can be performed easily.

### 3.5. Validation of Selected Models with VeVaPy 

Our demonstration of a validation procedure using novel TSST data and models from the literature illustrates how VeVaPy makes this process more streamlined. We started validation with VeVaPy by using parameter values found in the original publications of the selected models (see the model files on GitHub for details). 

With the authors’ published parameters and the data from patient 40, we see in Figure 6 that the original parameter values provided in Sriram et al. and data from this patient do not agree. When a new experimental procedure is used to collect data, it is likely that there will be no models specifically designed to simulate the experiment. This partially explains why the V&V process is more challenging in systems physiology. Since the data for model construction and data for model validation are often collected in different contexts, more than one set of parameter values are needed for proper estimation of the model’s capacity to explain new data. 

To maximize this capacity, VeVaPy includes a package for parameter optimization. This ensures that when the published parameters are inaccurate for the current experimental conditions, we can determine whether a change in parameters can yield a more accurate simulation. This is easily facilitated by VeVaPy, with several parameter optimization algorithms and cost function options easily available.

The optimized parameters for Sriram et al. improved the matching between the model and data from patient 40, as seen in Figure 7. The model fits exceptionally well when matching the data from patient 40, with an average cost function value over five iterations of the parameter optimization algorithm of 0.05827298. 

We see similar results when running the model by Bangsgaard et al. in VeVaPy against the data from patient 40. The initial results using the authors’ published parameters are shown in Figure 8, and as expected the simulation does a very poor job matching the experimental data.

After running the Bangsgaard model with VeVaPy’s parameter optimization function, we see significantly improved fit when matching the data from patient 40. As shown in Figure 9, the model performs nearly as well as the Sriram model in this instance.

For the remaining three models, the results of parameter optimization are not nearly as positive, but optimization still yielded slight improvements in fit. Figure 10 shows the models with and without parameter optimization against patient 40 (as with the models by Sriram et al. [23] and Bangsgaard & Ottesen [26] above). This process clearly demonstrates that even with parameter optimization, models are often not suitable for problems outside of their initial intended use.

For each model, we have computed the average cost function value for five iterations of the parameter optimization algorithm on each of the seven data sets tested. We then took the average of these seven average cost function values to obtain a single value representing the overall suitability of each model when matching our TSST data. The overall average cost function values given by VeVaPy for each model are shown in Table 2, alongside the best cost function value on a single patient, and the cost function values of the models without parameter optimization. It should be noted that the model by Bairagi et al. [22] required a large amount of computational power, and as such, we were only able to run one iteration of the parameter optimization algorithm for each data set. Each iteration of the model ran for approximately 36 h, which was more than 10 times longer than any of the other models we tested.

Based on the overall cost function value of each model with optimized parameters versus the authors’ published parameters, we can clearly see that our procedure is yielding significant improvements in data matching. Further, between the models tested, we see widely varying levels of suitability after parameter optimization. The models by Sriram et al. [23] and Bangsgaard et al. [26] far outperform the others. The normalization performed when computing costs allows for comparison between models without needing to convert time/concentration scales beforehand—we have reported results from each model without converting all models to the same scales to demonstrate this.

### 3.6. VeVaPy Facilitates Efficient Validation against Individual Patients 

With VeVaPy, we efficiently compared data from seven individual patient data sets against five models with five iterations per patient, and one model with a single iteration per patient, for a total of 182 runs of the parameter optimization. The main time-consuming step is the repeated integration of these models necessary for parameter optimization—especially for very complex models or those with systems of DDEs. However, VeVaPy makes it straightforward to “plug and play” individual models and data. The results of model validation against the MDD patient mean data set are shown for all five models in Figure 11. 

VeVaPy is designed to facilitate this process by requiring specification of the validation data set in a single location. The tool then runs the algorithm and outputs the results to an Excel file for further analysis. There is also the option in VeVaPy to loop over multiple patients in a data set, running the parameter optimization on each individual indicated by the user. This makes it very efficient to run many different validation tests, without needing to make changes to the code.

### 3.7. Assessing Model Generalizability

After optimizing parameters against a data set, VeVaPy can easily test the resulting parameter set against other data sets to determine whether a model can be generalized to other situations. The optimized parameters are loaded from the file where they were saved when generated and the procedure for performing a simulation without optimization is followed. In order to demonstrate this process, we have used several of the optimal parameter sets from the model by Sriram et al. [23] to run simulations against all individual patients from the TSST data set Figure 12 highlights some of the more interesting results from this process.

The results of these simulations vary widely, likely due to the variation observed between individual patients. Some parameter sets did yield better cost function values when matching individual patients within the same group. For instance, using parameters from optimization against all control patients gave an average cost function value of 6.007 against control patients and a value of 9.064 against MDD patients. Likewise, using parameters from optimization against patient 40 (MDD/neither subtype) gave an average cost function value of 2.899 against control patients and a value of 2.380 against MDD patients. However, using parameters from optimization against all MDD patients gave an average cost function value of 4.195 against control patients and a value of 4.421 against MDD patients. This demonstrates that while some optimized parameters are slightly generalizable to other patients or group mean concentrations, the cost function values are much higher than we would like in all of these situations. A deeper analysis of this issue falls outside of the scope of this paper, but we intend to examine this behavior more fully in a subsequent paper. 

### 3.8. VeVaPy Runtimes

We have recorded the time required to run a variety of simulations with each demonstration model. These data are summarized in Table 3 below. Running simulations without parameter optimization requires only milliseconds for all five models. However, optimizing parameters for any of the models requires a significant investment in time and computing power. The models vary widely in this regard, though, with a gap of 96.5 min between the fastest model and slowest model. 

The simulations performed for the average runtimes without parameter optimization used several optimal parameter sets that were generated during the course of model validation. We ran 100 simulations with each model, and the average runtime of these 100 simulations is reported. For the runtimes with parameter optimization, we report the average time for a single iteration of the optimization algorithm for each model against two data sets from the TSST data: mean of all control patients and mean of all MDD patients. Each of these optimization runs consisted of five iterations of the differential evolution optimization algorithm using a population size of 10. Population size determines how many parameter sets are “evolved” and checked for improvement at each step of the optimization, so lower population sizes will yield faster runtimes with less accuracy at finding the minimum cost value.

One major factor increasing the runtime for the models by Malek et al. [25] and Bairagi et al. [22] is the presence of delayed variables. This requires extra computation at each step of integration in order to look up the value of the variable in a previous time step, which becomes a significant time cost when it is being performed thousands of times per iteration of the optimization algorithm. This is an area where there is likely room for improvement in the VeVaPy framework—and one which we will be working on in the future.

## 4. Discussion

In this work, we have shown how we have created a Python based package, VeVaPy, that can be used to efficiently verify and validate HPA axis models. We have thoroughly documented the code behind VeVaPy and published it freely on GitHub, in line with the recommended best-practices for model publication. We hope that others will find VeVaPy useful, and it can help future researchers spend less time and effort performing V&V when developing their own models or checking model papers in the literature. 

In order to test and demonstrate VeVaPy, we verified five HPA axis models from the literature and validated them against novel TSST data from MDD patients. The models were ranked based on their average cost function value when running the differential evolution parameter optimization algorithm on each model against several TSST data sets. All five models are included in the VeVaPy repository and ready for use—though the validation results indicate that the models by Sriram et al. [23] and Bangsgaard & Ottesen [26] would be the strongest candidates for repurposing to explain the TSST data.

Consistent with many others in the scientific community, we have found that verification of published models was challenging [32,33,34,35]. As we will elaborate on below, we have encountered two main difficulties during the course of this research: data sets not provided alongside models, and non-standardized model development and publishing practices.

The first difficulty arises due to a lack of easily accessible data published in machine-readable formats. While many papers in the HPA axis modeling literature use cortisol concentration data to validate their models, they seldom include a Appendix A with a spreadsheet of the data used. Often, the papers cited by these modeling publications as the source of the data used do not include spreadsheets of the data either. It is often very difficult or even impossible to reach the authors of papers with useful data, especially when the papers are not from the last few years. 

To prevent future researchers from experiencing this challenge, we have begun the process of curating data sets from the literature and packaging them with VeVaPy. Currently, as described in Section 2.3, we have four data sets containing cortisol data sampled at short intervals from patients at rest over 24 h (two of the data sets also contain ACTH data over the same time period). These data are useful for validation of models intended to describe the HPA axis at rest. While the number of patients per data set is rather small (n = 29–47), and the sets only contain mean concentration data, they provide a basic level of confidence that a model is valid among various patient populations. Further, for validation of models intended to simulate the HPA axis under stress, the TSST data contained in VeVaPy is unique in the literature, as far as we are aware.

The second difficulty we faced—non-standardized model development and publishing practices—warrants a more in-depth discussion. As mentioned briefly in Section 2.1, there were many more models published in the literature than the five we have reproduced here. Unfortunately, however, the majority of differential equation based HPA axis models published in the literature are non-reproducible for a variety of reasons. The problem is primarily due to a lack of information, rather than dishonesty or poor model design/performance. Very few models are published with the full code used by the authors, and others do not include necessary basic information such as the initial conditions used for each simulation, or a full list of parameter values used.

It is encouraging that both some grant agencies and some journals have acknowledged this challenge. In a statement by the director of the NIH published in 2014 [36], the root causes of the crisis (including over-emphasis on high-impact journal publications by hiring and tenure committees or the withholding of information about experimental procedures to retain a competitive edge) were discussed along with the steps the NIH was considering to address the crisis (including changes to the way grants are awarded to allow for more reproduction of published work to take place). Meanwhile, other researchers have suggested that journals must enact and enforce reproducibility standards to solve the crisis [37,38]. Some journals have taken steps in the right direction—notably, Science implemented a reproducibility policy in February 2011. According to Stodden et al. [39], the policy of Science has been successful to a degree but has not been enforced strictly enough. The rate of data availability improved from 52% to 75%, but the rate of code availability only improved from 43% to 54% [39]. However, most suggestions for making computational research more reproducible focus on individual researchers.

A survey of 1576 scientists showed favorable attitudes towards all suggested practices for improving reproducibility included in the survey (practices such as better mentorship, more robust experimental design, and journal checklists) [40]. A concern expressed by some researchers surveyed was the amount of added time and effort to ensure that an experiment is reproducible. However, as stated by Waltemath & Wolkenhauer [41], “irreproducibility hinders researchers and the scientific community by wasting time and money.” Following best practices for reproducibility in computational research from the start of model development can decrease the overall amount of work required for reproducibility and in the long term it will save the community significant time and effort.

To ensure that VeVaPy is user friendly and easily extensible, we have followed the suggested best practices in the literature to the best of our ability—and we will review and summarize them here. We hope that by proposing (and following ourselves) these best practices, future computational biological scientists will not struggle with some of the challenges we have. The Physiologically Based Kinetic (PBK) Model Reporting Template presented by the Organisation for Economic Co-operation and Development (OECD) [32] is a good compilation of general best practices for model reproducibility. The template is presented in Table 4 below, and Table 5 contains our curated list of best practices suggested in the literature by various authors [32,33,34,35,37,38,41,42,43,44,45,46,47,48,49,50,51,52].

Note that the version of the OECD PBK Model Reporting Template published in [32] includes more thorough guidance about what to include in each section. Completing this template guarantees that a model will be published in accordance with our general best practices (see Table 5), assuming the authors also provide the full code of their model as the template suggests. See the Appendix A for complete information on the five models covered in this paper.

We can categorize best practices suggestions based on which aspect of an experiment they address. The categories include experimental design, performing experiments and collecting data, analysis of data, and reporting data/results. The paper by Munafò et al. [45] presents general suggestions which are applicable to many areas of science, and which address all of the aforementioned categories. These suggestions include protecting against cognitive bias during experimental design and data collection (e.g., using blinding), including independent researchers with no personal stake in all steps of an experiment, study pre-registration, improving statistical analysis training, improving the quality of reporting, and promoting transparency and open science.

Due to the nature of computational research, the suggestions regarding experimental design and data collection are often not relevant. As such, the literature about reproducibility in computational science mostly focuses on the last category listed above: reporting data/results. The suggestions in this category vary in their specificity from general statements (see the suggestions listed in Table 5) to specific software recommendations (e.g., use Git for version control).

We have also made a list of suggested best practices and the software we recommend for implementing them (see Table 6). These suggestions come from several literature sources and our own experience with modeling software [33,41,42,46,47,48,49,50,52]. The following best practices and the suggested software for implementation will be discussed: fully document the process of model development including all simulation inputs and algorithms, share model code and the associated documentation in public repositories like Github, ensure that model code can be run on as many computers as possible, and make model code easy to understand.

In our experience and that of Kim et al. [42], documentation of the model development process including all simulation inputs and algorithms is most easily accomplished using computational notebooks and version-control systems. The code included with this paper is written in Python using Jupyter notebooks, as explained in Section 2.2. The version-control system we utilized is called Git, which allows for users to save all versions of a program from its creation. This allows for easily stepping back through versions to see when a change was made or to determine when an error was introduced. Through using these tools in tandem, we have fully documented the development process of VeVaPy.

Using Git for version-control makes it simple to deposit model code in a public, version-controlled repository—GitHub is a website which allows for any Git repository to be uploaded to the Internet and (optionally) made public. This is the option that we have chosen to use, and VeVaPy can all be found at https://www.github.com/cparker-uc/VeVaPy (accessed 23 November 2022). However, there are also specific repositories for various types of models. Porubsky et al. [52] recommend BioModels (https://www.ebi.ac.uk/biomodels/, accessed 23 November 2022), a database for biological models that has a curation process which verifies whether uploaded models are reproducible.

Due to the difficulties in ensuring that future users have all requisite software installed in the correct versions, even when all code is included with a publication or deposited in an open repository it can prove difficult to run—especially when the code is not from the past couple of years. To ensure that model code can be run on as many computers as possible, Porubsky et al. [52], Sandve et al. [46], Waltemath & Wolkenhauer [41], and Rule et al. [49] suggest using either a virtual machine image or a web-based virtual machine such as Docker (https://www.docker.com, accessed 23 November 2022). Using a virtual machine, one can be sure that all necessary software, data, and model code will be available and able to run in any computing environment. We have instead opted for Binder (https://www.mybinder.org, accessed 23 November 2022) which allows users to run Jupyter notebooks in a web browser without needing to have the necessary software installed on their local machine.

The final suggestion we will discuss is making model code easy to understand. This can be achieved with thorough documentation, use of compartmentalization and functional programming, descriptive variable names, and using computational notebooks. However, for certain forms of model, there are standardized markup languages which are widely recommended in the literature, and which make model code easier for users to understand [33,41,48,50,52]. For systems biology, these include the systems biology markup language (SBML) and CellML. Unfortunately, as discussed in Medley et al. [38], these languages are limited in scope and do not support all forms of systems biology modeling. For instance, the models covered in this paper are currently unable to be adapted to SBML or CellML. However, we felt that it was important to mention these languages as they offer significant upsides for those models which they support.

VeVaPy followed biological modeling best practices as discussed above (Table 5), and this has made the five models reproduced for this research and our code, VeVaPy, very useful for many HPA axis modeling applications. To summarize, we have made the following efforts: the code for our VeVaPy package includes thorough documentation, including instructions for use; the demonstration models were implemented using Jupyter notebooks for improved readability and easier documentation; we have tracked the development process with the Git version control system and published it on GitHub, a freely accessible code repository; and we have provided instructions for accessing VeVaPy through Binder to facilitate its use on any Internet-connected computer.

We intend to follow up on the models used here for demonstration and to repurpose the model by Sriram et al. [23] to specifically match data that includes acute stressors (such as the TSST data used above). The most apparent modification to be made involves replacing the variable for stress input to a function for stress input that can change over the course of the simulation. This will allow for the introduction of an acute stressor (like a TSST) and then the cessation of the stressor afterwards. Another modification that we will test is the addition of delays between the release of ACTH and its action in the adrenal glands and between the production of cortisol and its feedback in the hypothalamus and pituitary. There are many possible routes that our modifications may take due to the many physiological processes not yet accounted for in the literature models (GRs in the hippocampus, fast vs. slow cortisol feedback, etc.). Using a model specifically designed to simulate stress tests will allow us to better understand the behavior of the HPA axis under acute stress—and the differences in this behavior between MDD, PTSD, and healthy control subjects. 

Although we are hopeful that future published models will be more easily reproducible and include all data used by the authors for validation, we believe that the reproducibility problems discussed above can be eased to a degree by the development of robust tools for model V&V. We have begun the development of such tools in the field of HPA axis modeling. Although VeVaPy currently requires some programming knowledge to adapt beyond the five included models, we intend to further develop it into a graphical user interface for easily creating systems biology and systems pharmacology models. This will hopefully allow biologists with no experience in modeling to use our tools on many published mathematical models and improve the reach, and therefore the power to make an impact, of mathematical modeling in general. It is our firmly held belief that the development of tools to facilitate V&V of mathematical models, such as VeVaPy, will speed the pace of research and ensure that identification of valid models in the literature takes significantly less effort.

## Figures and Tables

**Figure 1 entropy-24-01747-f001:**
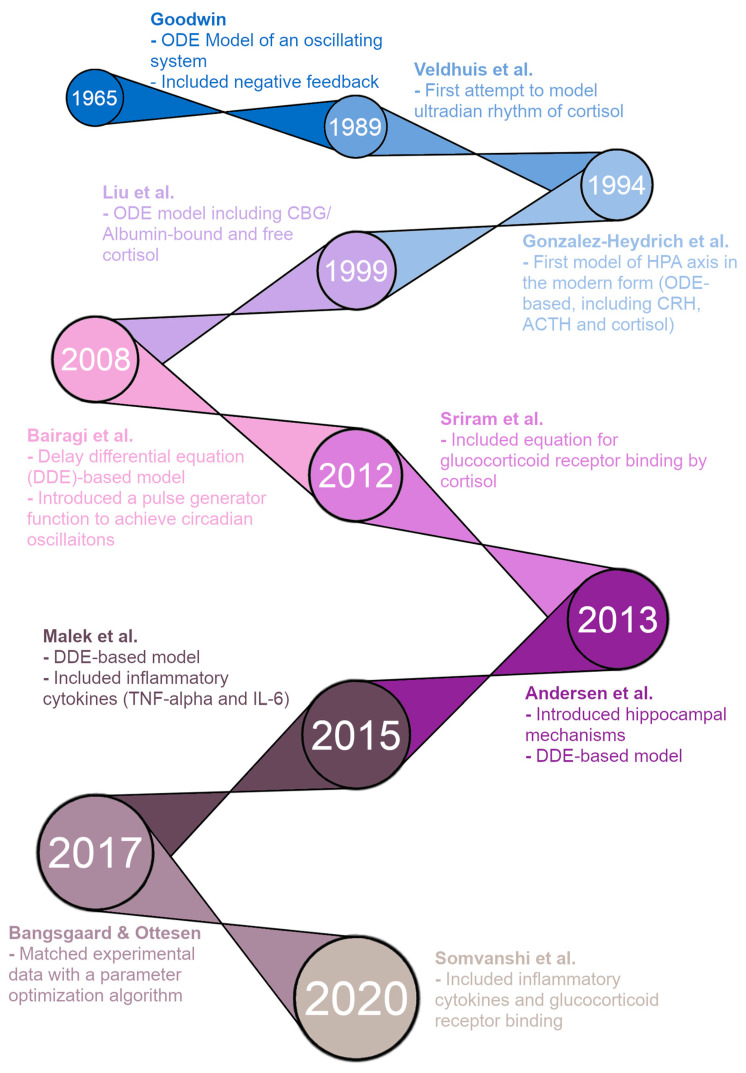
Timeline of hypothalamic-pituitary-adrenal (HPA) axis modeling. Models included, in chronological order: Goodwin [18], Veldhuis et al. [19], Gonzalez-Heydrich et al. [20], Liu et al. [21], Bairagi et al. [22], Sriram et al. [23], Andersen et al. [24], Malek et al. [25], Bangsgaard & Ottesen [26], and Somvanshi et al. [27].

**Figure 2 entropy-24-01747-f002:**
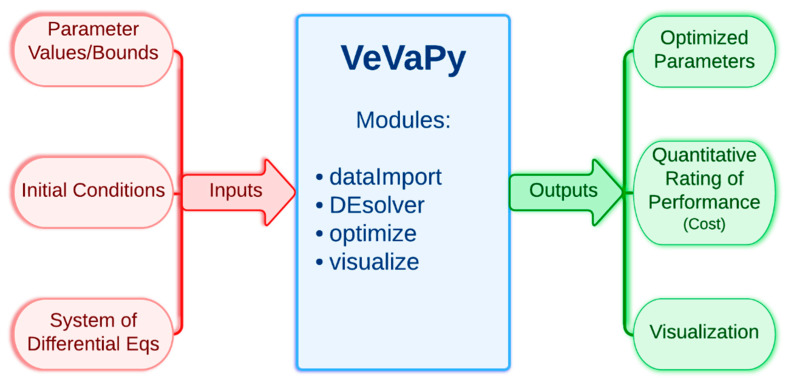
Simplified diagram of VeVaPy, showing inputs and outputs of the tool.

**Figure 3 entropy-24-01747-f003:**
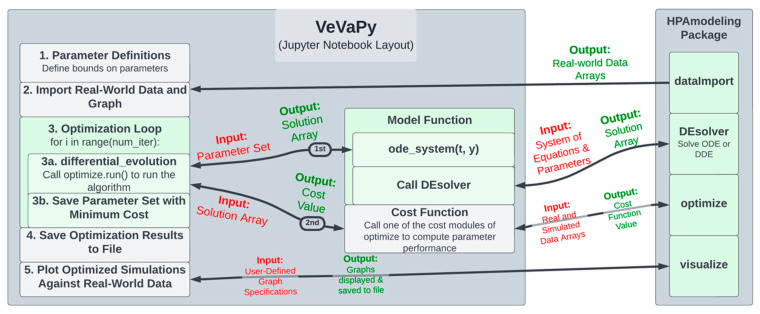
Code diagram of VeVaPy.

**Figure 4 entropy-24-01747-f004:**
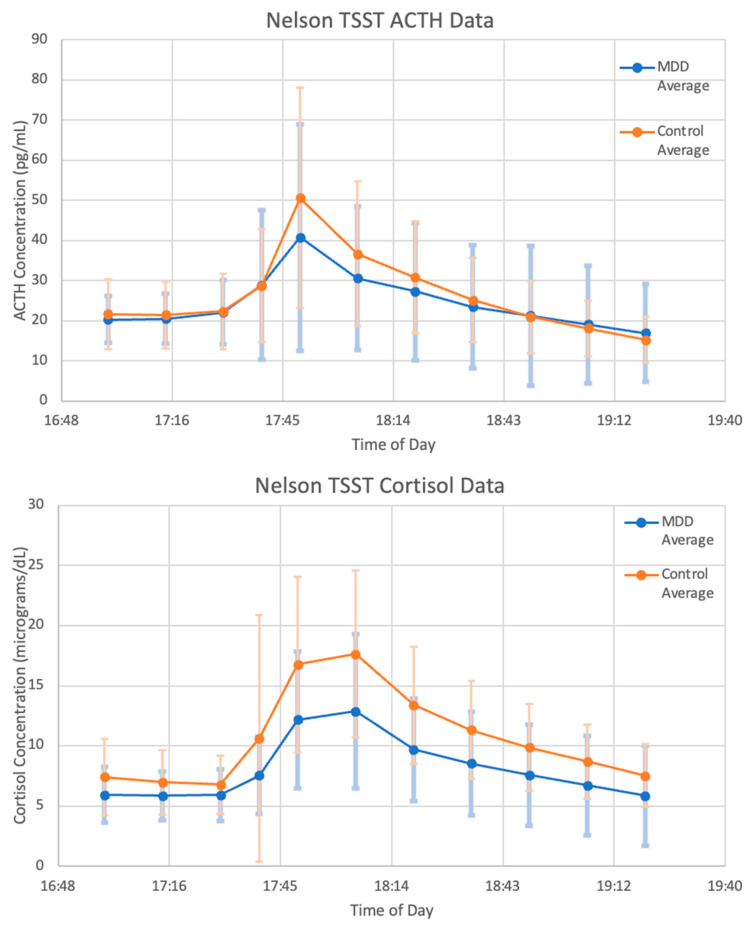
Mean of MDD (blue) and Control (orange) Patients’ Trier social stress test (TSST) data. Adrenocorticotropic hormone (ACTH) concentration (**top**) and cortisol concentration (**bottom**) are graphed without any model simulations. Vertical bars indicate standard deviation at each data point.

**Figure 5 entropy-24-01747-f005:**
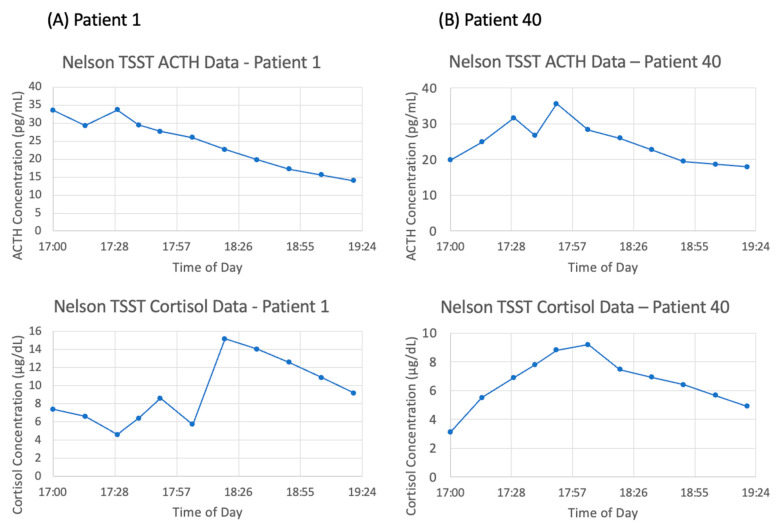
Trier social stress test (TSST) data from (**A**) patient 1 and (**B**) patient 40. Adrenocorticotropic hormone (ACTH) concentrations (**top**) and cortisol concentrations (**bottom**) are graphed without any model simulations.

**Figure 6 entropy-24-01747-f006:**
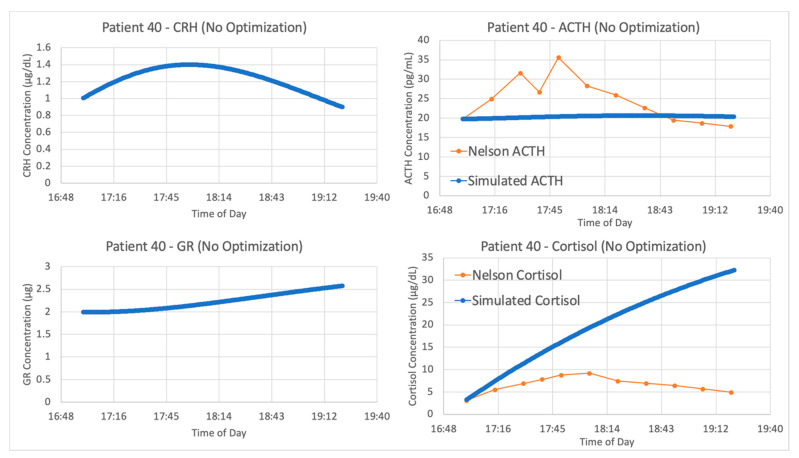
Sriram et al. [23] model without parameter optimization vs. Trier social stress test (TSST) data from patient 40. Graphs include model simulations of corticotropin-releasing hormone (CRH) concentration (**upper left**, blue), adrenocorticotropic hormone (ACTH) concentration (**upper right**, blue), cortisol concentration (**lower right**, blue) and bound glucocorticoid receptor (GR) concentration (**lower left**, blue) against ACTH concentration (**upper right**, orange) and cortisol concentration (**lower right**, orange) from patient 40. The blue lines represent the average of 5 iterations of the parameter optimization algorithm.

**Figure 7 entropy-24-01747-f007:**
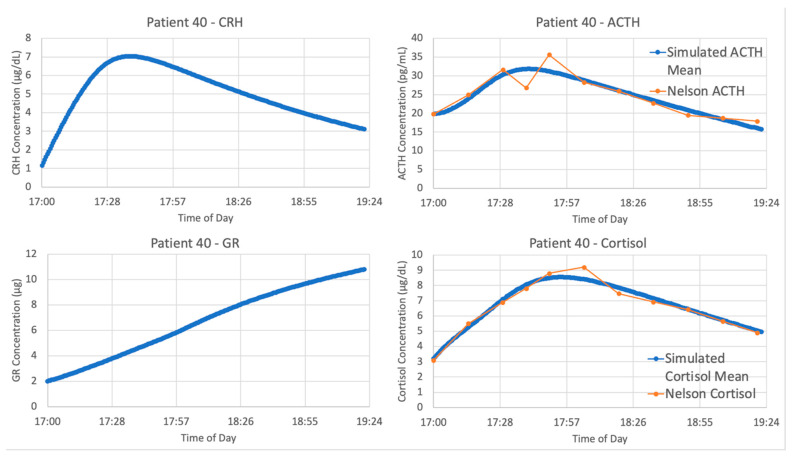
Sriram et al. [23] model vs. Trier social stress test (TSST) data from patient 40. Graphs include model simulations of corticotropin-releasing hormone (CRH) concentration (**upper left**, blue), adrenocorticotropic hormone (ACTH) concentration (**upper right**, blue), cortisol concentration (**lower right**, blue) and bound glucocorticoid receptor (GR) concentration (**lower left**, blue) against ACTH concentration (**upper right**, orange) and cortisol concentration (**lower right**, orange) from patient 40. The blue lines represent the average of 5 iterations of the parameter optimization algorithm.

**Figure 8 entropy-24-01747-f008:**
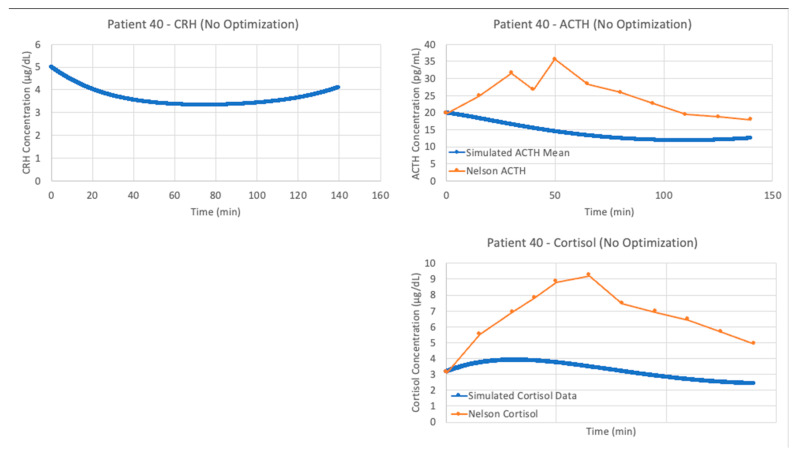
Bangsgaard et al. [26] model without parameter optimization vs. Trier social stress test (TSST) data from patient 40. Graphs include model simulations of corticotropin-releasing hormone (CRH) concentration (**left**, blue), adrenocorticotropic hormone (ACTH) concentration (**upper right**, blue) and cortisol concentration (**lower right**, blue) against ACTH concentration (**upper right**, orange) and cortisol concentration (**lower right**, orange) from the patient 40. The blue lines represent the average of 5 iterations of the parameter optimization algorithm.

**Figure 9 entropy-24-01747-f009:**
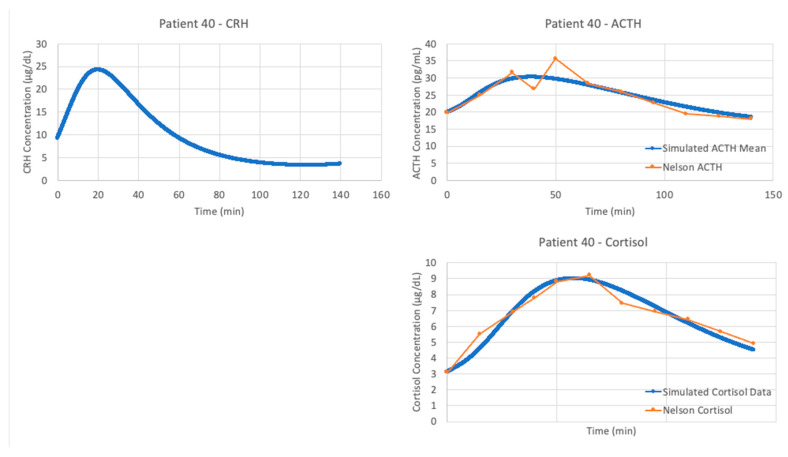
Bangsgaard et al. [26] model vs. Trier social stress test (TSST) data from patient 40. Graphs include model simulations of corticotropin-releasing hormone (CRH) concentration (**left**, blue), adrenocorticotropic hormone (ACTH) concentration (**upper right**, blue) and cortisol concentration (**lower right**, blue) against ACTH concentration (**upper right**, orange) and cortisol concentration (**lower right**, orange) from the patient 40. The blue lines represent the average of 5 iterations of the parameter optimization algorithm.

**Figure 10 entropy-24-01747-f010:**
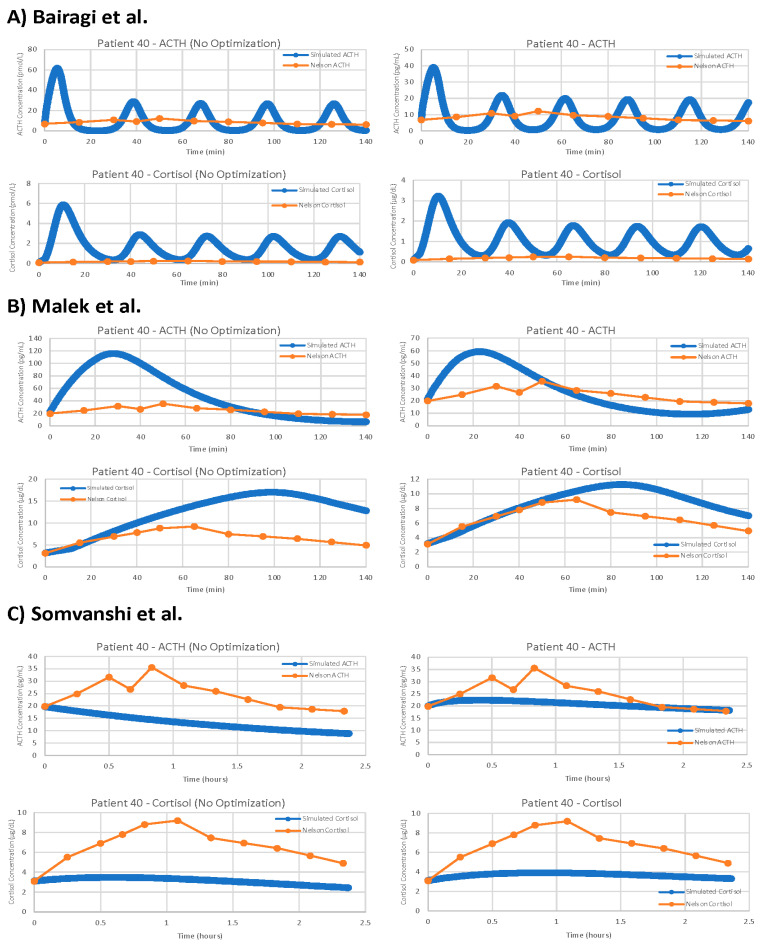
Further examples of models with and without parameter optimization. Simulated concentrations are in blue, patient 40 data is in orange. The left column of graphs shows the models running simulations with the parameters from the publication, while the right column of graphs shows the models running simulations with optimized parameters. Demonstrated models are (**A**) Bairagi et al. [22], (**B**) Malek et al. [25], (**C**) Somvanshi et al. [27].

**Figure 11 entropy-24-01747-f011:**
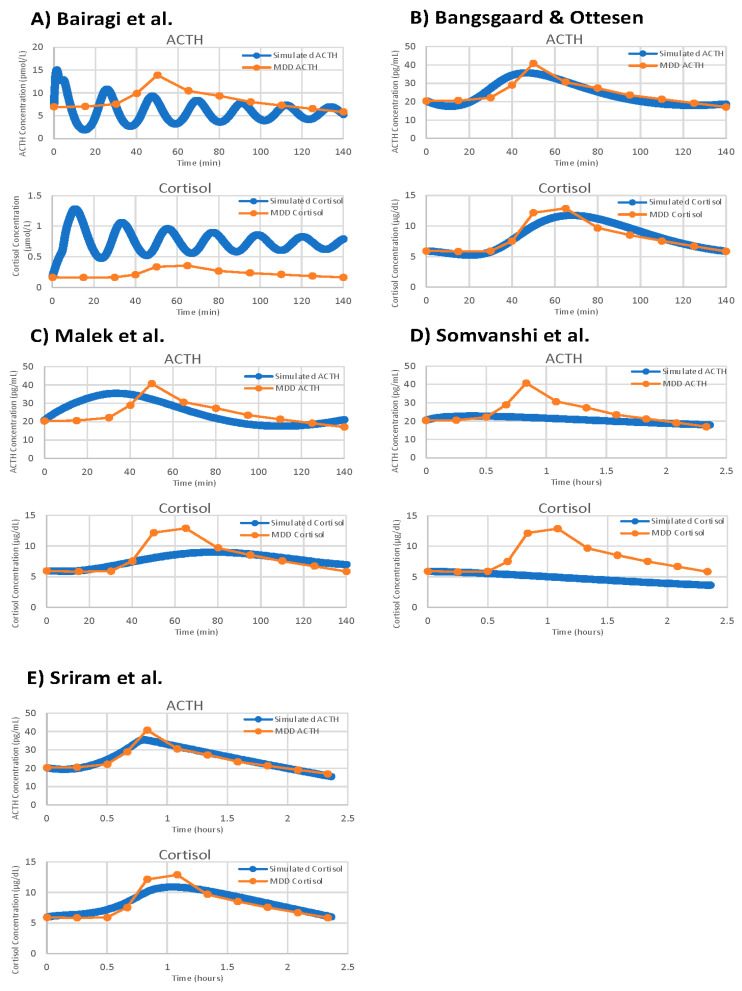
Model validation figures for all five demonstration models against the mean of all MDD patients in the Nelson TSST data. Models depicted are: (**A**) Bairagi et al. [22], (**B**) Bangsgaard & Ottesen [26], (**C**) Malek et al. [25], (**D**) Somvanshi et al. [27], (**E**) Sriram et al. [23].

**Figure 12 entropy-24-01747-f012:**
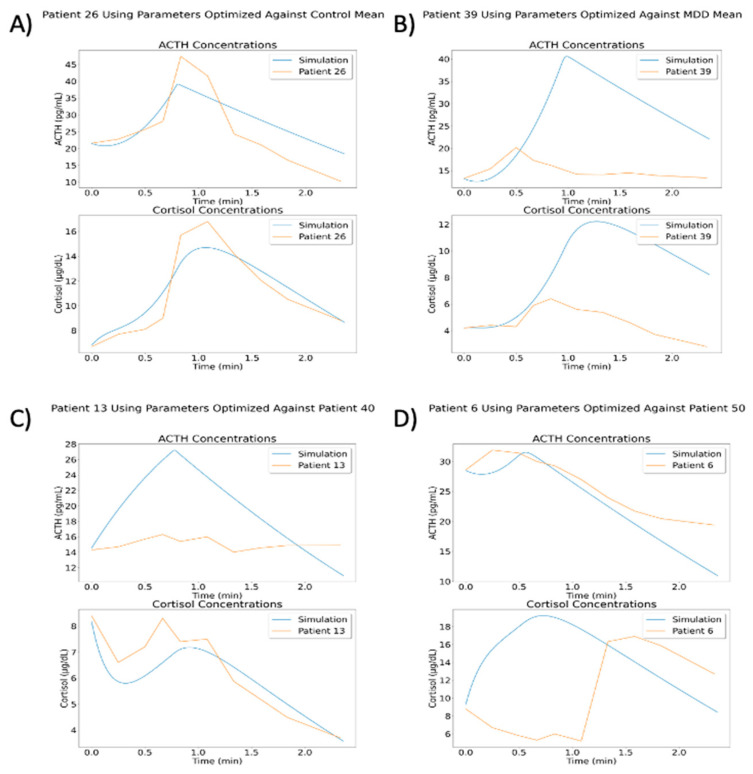
Results of Using Optimized Parameters in Generalized Cases. (**A**) The optimized parameter sets have some cases where they perform reasonably well (especially against patients from the same group). (**B**) Some of the parameter sets match certain patients very poorly, such as the parameters optimized against the mean of all MDD patients against patient 39 (MDD/neither subtype). (**C**) Many of the simulations matched either ACTH or cortisol but did not match the other. Parameters optimized against patient 40 (MDD/neither subtype) match the general cortisol concentration trend from patient 13 (MDD/atypical), but the simulated ACTH concentration is extremely high. (**D**) Similar to C, but with simulated cortisol concentration not matching while simulated ACTH concentration follows the correct general trend. Simulation run with parameters optimized against patient 50 (MDD/atypical), shown with data from patient 6 (MDD/atypical).

**Table 1 entropy-24-01747-t001:** Overview of Selected Models.

Year	Authors	Target	Number of Equations	Number of Parameters	Number of Feedback Loops	Unique Features
2008	Bairagi, Chatterjee, Chattopadhyay	Circadian & Ultradian Rhythms	3	12	1 Negative	Ultradian rhythm w/o Circadian
2012	Sriram, Rodriguez-Fernandez, Doyle	Cortisol Levels in PTSD	4	20	1 Negative, 1 Positive	Glucocorticoid Receptor Binding
2015	Malek, Ebadzadeh, Safabakhsh, Razavi, Zaringhalam	HPA Axis Relationship to Inflammatory Cytokines	5	32	0	Equations for TNF-alpha, IL-6 and endotoxin
2017	Bangsgaard, Ottesen	Comparing Differences in Optimal Parameters between Individuals	3	17	1 Negative	Used Parameter Optimization on Model to Compare Individuals
2020	Somvanshi, Mellon, Yehuda, Flory, Bierer, Makotkine, Marmar, Jett, Doyle	Relation of HPA Axis to Inflammation in Subjects with PTSD	17	92	1 Negative, 1 Positive	More Detailed Glucocorticoid Receptor and Inflammatory Cytokine Dynamics

**Table 2 entropy-24-01747-t002:** Model Ranking Based on Cost Function Value.

Model	Overall Cost Function Value ± Standard Deviation (After Parameter Optimization)	Best Cost Function Value for a Single Patient (Patient ID)	Overall Cost Function Value (Authors’ Parameters, No Optimization)	Best Cost Function Value for Single Patient (Patient ID) (Authors’ Parameters, No Optimization)
Sriram et al. (2012) [23]	0.33 ± 0.19	0.058 (Patient 40)	26.03 ± 16.39	5.11 (Patient 50)
Bangsgaard & Ottesen (2017) [26]	0.58 ± 0.46	0.12 (Patient 40)	3.63 ± 1.03	2.14 (Patient 10)
Somvanshi et al. (2020) [27]	2.59 ± 0.94	1.10 (Patient 20)	6.61 ± 0.90	5.43 (Patient 20)
Malek et al. (2015) [25]	6.78 ± 7.78	1.17 (Patient 30)	11.95 ± 0.40	11.31 (Patient 20)
Bairagi et al. (2008) [22]	64.86 ± 38.85	35.84 (Patient 1)	656.26 ± 280.47	343.34 (Patient 1)

**Table 3 entropy-24-01747-t003:** Simulation Runtimes with VeVaPy by Model.

Model	Without Optimization (Milliseconds)	With Optimization (Minutes)
Bairagi et al. [22]	483	101.625
Bangsgaard & Ottesen [26]	51.1	27.170
Malek et al. [25]	57.5	26.794
Somvanshi et al. [27]	5.1	5.128
Sriram et al. [23]	1.76	7.159

**Table 4 entropy-24-01747-t004:** OECD Model Reporting Template [32].

PBK Model Reporting Template Sections
A. Name of model
B. Model author and contact details
C. Summary of model characterization, development, validation and regulatory applicability
D. Model characterization 1. Scope and purpose of the model 2. Model conceptualization (model structure, mathematical formulation) 3. Model parameterization (parameter estimation and analysis) 4. Computer implementation (solving the equations) 5. Model performance 6. Model documentation
E. Identification of uncertainties (report for each item in D.)
F. Model implementation details (software used, availability of code)
G. Peer engagement (report extent of review by peers during development)
H. Parameter tables (report all relevant inputs to the model for any simulations described)
I. References and background information

**Table 5 entropy-24-01747-t005:** Proposed Best Practices for Model Publication.

Proposed Best Practices	Comments/Justification
All models should be published with all code used by the authors	This is essential to ensure models can be exactly reproduced without undue struggle
Model code must include proper documentation	Without documentation regarding how to run a model, such as thorough comments throughout the code or a readme file, it is often difficult to dissect complex code and determine how it is meant to work
Exact scope of the model should be made clear	It is important that the audience knows when it is appropriate to use the model, lest they form false assumptions based on use of the model in a context it was not designed to simulate
All input data (parameters, initial conditions, etc.) must be provided and justified	Too often models are published without a clear list of parameter and initial condition values, making them non-reproducible. Other times, the sources for parameter and initial condition values are not provided, leaving their validity in question.

**Table 6 entropy-24-01747-t006:** Best practices and suggested software for their implementation.

Best Practice Suggestion	Software for Implementation
Fully document the process of model development including all simulation inputs and algorithms	Jupyter Notebooks (https://www.jupyter.org, accessed 23 November 2022) R Notebooks (https://rmarkdown.rstudio.com, accessed 23 November 2022)
Share model code and the associated documentation publicly	GitHub (https://www.github.com, accessed 23 November 2022) BioModels (https://www.ebi.ac.uk/biomodels, accessed 23 November 2022)
Ensure that model code can be run on as many computers as possible	VMWare Virtual Machine (https://www.vmware.com, accessed 23 November 2022) Docker (https://www.docker.com, accessed 23 November 2022) Binder (https://www.mybinder.org, accessed 23 November 2022)
Make model code easy to understand	Systems Biology Markup Language (SBML, https://synonym.caltech.edu, accessed 23 November 2022) CellML (https://www.cellml.org, accessed 23 November 2022)

## Data Availability

All data used in the study has been made available in the Appendix A.

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
