# Peer review of "VeVaPy, a Python Platform for Efficient Verification and Validation of Systems Biology Models with Demonstrations Using Hypothalamic-Pituitary-Adrenal Axis Models"

_entropy, 2022, doi:10.3390/e24121747_

Round 1
Reviewer 1 Report
This study develops a platform for verification and validation of HPA axis models. Five HPA axis models are selected and their performance are compared by using data of subjects who underwent a trier social stress test (TSST). The platform can save time for future researchers to select HPA axis for their application. The paper is well written and I have a few comments that follow.
1) You have a rich data set (58 subjects). Due to computational time, a subset of them are used for parameter estimation. Have you checked the performance of the model for other subjects not used in the calibration process? (set the initial values of ACTH and Cortisol for such patients (not used in parameter estimation) and check the performance of the model with the parameters that you optimized).
2) The explanation about Model Validation on page 5 is a bit confusing. At the end of the first paragraph, it is mentioned “For Validation, we run a parameter optimization algorithm on each model to determine optimal parameters for matching stress test data form seven patients and the mean concentrations of all patients”. This sentence implies that validation is parameter estimation. I suggest to move this sentence to the next paragraph in which you explain comparison of the model output and data.
3) Why does the cost function involve creating splines between simulated points for the hormones? Why not comparing simulated data points and the real data points?
4) In the result section on page 12, do you mean figure 4 (not 2)? The same comment for the next page (see figure 5, not 3)?
Reviewer 2 Report
The manuscript titled “VeVaPy: A python platform for efficient verification and validation of systems biology models with demonstrations using HPA axis models” summarizes their efforts towards the standardization of the systems biology models for the purpose of efficient verification and validation. The VeVaPy platform is developed and used to demonstrate the usability of the five HPA-axis models from the previous publication along with the parameter optimization and validation on a different set of clinical data (TSST data from the MDD patient cohort). I have following comments to the authors:
1- As the paper involves parameter optimization as one of the core contribution, it would be better if the mathematical equations related to the cost function in the optimization algorithm are elaborated in the manuscript.
2- It is not clear how each model from the literature was used to generate the profiles for TSST data. The mathematical equations that were additionally added/ modified in the existing models apart from the parameter optimization needs to be reported for better clarity of the methodology.
3- In the model validation section, from Fig.6-Fig.9 authors have represented the model performance with and without parameter optimization for only three models. It can be suggested to include the performance of all the five models in similar fashion. These figures may be clubbed together in one figure with several subfigures. Although the model performances are tabulated, the graphical representation will be much consistent.
4-In case of the lower performing models even after parameter optimization, authors can provide analysis or comments on the discrepancy of the model fit to the TSST data.
5- In Figure 10 the axis labels for all the figures are not uniform (some are in hours and some in minutes). The time axis for all the figures should be represented either in minute to be consistent with other figures in the manuscript. It is also not clear whether the plots in Fig.10 were generated with the model with optimized parameters or the native parameters of the model. Figure 10.D can be rechecked as the model by Somvanshi et. al. (2020) does not show stress response although its cost function values are lesser than Malek et al.(2015), and Bairagi et al. (2008) where at least some stress response is noticed.
6- It will be helpful to have reported the additional modelling performed by authors to simulate stress response in each case. It may be noted that the time dimensions or units of parameters may vary from model to model; therefore the units of the stress function should be adopted appropriately to simulate the stress response. There is some mention about what was done in case of the model by Sriram et. al. in the discussion section, however a mathematical equations would be helpful for the readers.
Reviewer 3 Report
Please find attached my review of the paper. There are a few minor concerns listed, as well as an error report generated when I tried to open VeVaPy through Binder.
